# Circulating Monocytes Serve as Novel Prognostic Biomarker in Pancreatic Ductal Adenocarcinoma Patients

**DOI:** 10.3390/cancers15020363

**Published:** 2023-01-05

**Authors:** Frederik J. Hansen, Paul David, Marina Akram, Samuel Knoedler, Anke Mittelstädt, Susanne Merkel, Malgorzata J. Podolska, Izabela Swierzy, Lotta Roßdeutsch, Bettina Klösch, Dina Kouhestani, Anna Anthuber, Alan Bénard, Maximilian Brunner, Christian Krautz, Robert Grützmann, Georg F. Weber

**Affiliations:** 1Department of General and Visceral Surgery, Friedrich-Alexander-University, Krankenhausstraße 12, 91054 Erlangen, Germany; 2Institute of Regenerative Biology and Medicine, Helmholtz Center Munich, Ingolstädter Landtsraße 1, 85764 Neuherberg, Germany

**Keywords:** monocytes, pancreatic ductal adenocarcinoma, grading, perineural invasion, prognostic biomarker, liquid biopsy

## Abstract

**Simple Summary:**

Pancreatic ductal adenocarcinoma (PDAC) is one of the most lethal malignant diseases. The association of monocytes with worse outcomes has been demonstrated for a wide array of malignancies. Yet, their role in PDAC remains to be elucidated. In this study, we found elevated levels of circulating monocytes in PDAC patients, correlating with more aggressive tumor growth and decreased survival. Therefore, we propose that monocytes may act as a novel prognostic biomarker for PDAC. Future large-scale studies are needed to investigate monocytes as potential therapeutic targets.

**Abstract:**

Pancreatic ductal adenocarcinoma (PDAC) ranks among the most fatal cancer diseases, widely accepted to have the most dismal prognoses. Although immunotherapy has broadly revolutionized cancer treatment, its value in PDAC appears to be relatively low. Exhibiting protumoral effects, monocytes have recently been proposed as potential targets of such immunotherapeutic regimens. However, to date, the body of evidence on monocytes’ role in PDAC is scarce. Therefore, we analyzed monocytes in the peripheral blood of 58 PDAC patients prior to surgery and compared them to healthy individuals. PDAC patients showed increased levels of monocytes when compared to healthy controls In addition, patients with perineural infiltration demonstrated a higher percentage of monocytes compared to non-infiltrating tumors and PDAC G3 was associated with higher monocyte levels than PDAC G2. Patients with monocyte levels > 5% were found to have an 8.9-fold increased risk for a G3 and perineural infiltrated PDAC resulting in poorer survival compared to patients with <5% monocyte levels. Furthermore, PDAC patients showed increased expressions of CD86 and CD11c and decreased expressions of PD-L1 on monocytes compared to healthy individuals. Finally, levels of monocytes correlated positively with concentrations of IL-6 and TNF-α in plasma of PDAC patients. Based on our findings, we propose monocytes as a novel prognostic biomarker. Large-scale studies are needed to further decipher the role of monocytes in PDAC and investigate their potential as therapeutic targets.

## 1. Introduction

Pancreatic ductal adenocarcinoma (PDAC) is the seventh leading cause of cancer-related deaths worldwide with over 400,000 related deaths per year. More than 450,000 patients are diagnosed each year, with a generally negative expected prognoses [1]. The aggressive tumor growth of PDAC is reflected in one and five-year survival rates of 24% and 6%, respectively, and a median overall survival of less than four months [2,3,4]. In fact, radical surgical resection represents the only treatment modality with a potentially curative outcome [5]. However, owing to early metastatic dissemination, and late diagnosis, such operability remains rare: At the time of presentation, less than 25–30% of all patients may be considered candidates for (partial) pancreatectomy [6].

In locally advanced PDAC (LA-PDAC), the therapeutic approach is focused on local tumor and symptom control. Accordingly, LA-PDAC patients are commonly pretreated with neoadjuvant therapy, and resection is evaluated in case of response. In absence of sufficient response, the patient is tretated with palliative intent [7]. The preferred regimens in the neoadjuvant setting, the first line therapy for LA-PDAC are the FOLFIRINOX scheme (folinic acid, fluorouracil, irinotecan, oxaliplatin) or gemcitabine + nab-paclitaxel (GnP) [8]. Of note, the application of radiochemotherapy is being investigated in several studies. There is initial evidence that some patients benefit [9]. For metastatic PDAC, first-line therapy is stratified primarily by the patient performance status (ECOG) [10]: While ECOG < 1 patients can be treated aggressively with the FOLFIRINOX scheme, Gemcitabine-based regimens are indicated for patients with moderately impaired performance (ECOG 1-2). In cases of higher-grade ECOG, current guidelines advocate the strategy of best supportive care. Generally, along with life-prolonging measures, palliative-supportive therapy should be evaluated as early as possible and repeatedly [11].

Currently, joint research efforts are seeking to develop novel breakthrough therapies for PDAC. Although immunotherapy has revolutionized cancer treatment, its value in PDAC appears to be relatively low. Despite a plethora of clinical trials, to date only the use of anti-programmed cell death 1 antibodies for the treatment of metastatic PDAC with high microsatellite instability or DNA mismatch repair deficiency has been approved by the Food and Drug Administration [12].

PDAC evades immunological recognition by a variety of mechanisms, such as the induction of an immunosuppressive microenvironment [13,14]. In this context, monocytes emerged as heterogenous cells holding a regulatory role. These mononuclear phagocytes represent an important cell population of the innate immune system in the blood stream and the tumor microenvironment, with ambiguous oncosuppressive functioning [15]. In fact, recent studies report protumoral properties of monocytes, including the differentiation into tumor-associated macrophages, the promotion of cancerous angiogenesis, and the facilitation of metastatic cell seeding [16,17,18,19,20,21,22].

However, the body of evidence on monocytes‘ specific capacities in PDAC is lacking. Little is known about monocytes as PDAC diagnostics and therapeutic targets. This knowledge gap is exacerbated by the prediction that pancreatic cancer will be the second leading cause of malignant death by 2030 [23]. Therefore, in this study, we aimed to delineate the role of monocytes in PDAC.

## 2. Materials and Methods

### 2.1. Patient Samples

All patients aged 18 years or older with a postoperative histopathological diagnosis of pancreatic ductal adenocarcinoma (PDAC) or pancreatitis who underwent elective surgery at the Department of Surgery of University Hospital Erlangen, Germany, between 2020 and 2022 were eligible for inclusion in the present study. Peripheral blood was collected from 24 healthy individuals and 58 PDAC and 24 pancreatitis patients prior to surgery. The clinicopathological characteristics of the PDAC cohort are shown in Table 1.

### 2.2. Sample Preparation

The blood samples were collected in 7.5mL EDTA tubes (Cat-No. 04.1921.001, Sarstedt, Nürnbrecht, Germany). First, plasma was separated at a centrifugation of 350× *g* for 10 min without braking. Afterwards, 30 mL of 1× erythrocyte lysis buffer (Cat-No. 555899, BD Biosciences, Franklin Lakes, NJ, USA) was added to the cells followed by 15 min incubation at room temperature. Next, the cells were then centrifuged at 350× *g* for 5 min and re-suspended in 50 mL PBS (Cat-No. 14190169, Gibco, Waltham, MA, USA). To determine the absolute cell numbers, leukocytes were counted with trypan blue under the microscope. Finally, the cell suspensions were centrifuged at 350× *g* for 5 min at 4 °C, and the cell concentration was adjusted to maximum 1 million cells per 100 μL in PBS containing 1% FBS (Cat-No. A3160802. Gibco, Waltham, MA, USA), 0.5% BSA (Cat-No. A2153, Sigma-Aldrich, St. Louis, MO, USA), and 2 mM EDTA (Cat-No. AM9260G, Invitrogen, Waltham, MA, USA) (FACS buffer) for flow cytometric analyses.

### 2.3. Flow Cytometry

The following antibodies were used for flow cytometric analyses: anti-CD45-BV786 (Cat-No. 563716, BD Biosciences, Franklin Lakes, NJ, USA), anti-HLADR-BUV395 (Cat-No. 565972, BD Biosciences, Franklin Lakes, NJ, USA), anti-CD11c-BV711 (Cat-No. 563130, BD Biosciences, Franklin Lakes, NJ, USA), anti-CD14-BUV737 (Cat-No. 612763, BD Biosciences, Franklin Lakes, NJ, USA), anti-CD16-FITC (Cat-No. 406555, BD Biosciences, Franklin Lakes, NJ, USA), anti-CD86-BV510 (Cat-No. 563460, BD Biosciences, Franklin Lakes, NJ, USA), anti-PD-L1-BV650 (Cat-No. 563740, BD Biosciences, Franklin Lakes, NJ, USA). Data were acquired on a Celesta (BD Biosciences, Franklin Lakes, NJ, USA) flow cytometer using the BD FACSDiVa™ software v8.0.1.1 and analyzed with FlowJo 10.3.0 (FlowJo LLC, Ashland, OR, USA).

### 2.4. Plasma Cytokine Analysis

Plasma was separated from peripheral blood at a centrifugation of 350× *g* for 10 min without braking. Serum cytokines were analyzed by LEGENDPlex^TM^ bead-based immunoassays (Cat-No. 740527, Biolegend, San Diego, CA, USA) according to the manufacturer’s instructions. TNF-α, IL-6 and IL-10 were simultaneously quantified. Data acquisition was performed on flow cytometer and analyzed with the LEGENDPlexTM Data Analysis Software (Biolegend, San Diego, CA, USA).

### 2.5. Statistical Analysis

Data were gathered and saved in an electronic laboratory notebook and evaluated using GraphPad Prism (V9.00 for macOS, GraphPad Software, La Jolla, CA, USA). Continuous variables were analyzed with independent *t*-tests. To determine statistically significant differences between three or more groups, a one-way ANOVA test was applied. A Pearson’s Chi square was used to measure differences in categorical variables. Survival data were analyzed using the Log-rank (Mantel-Cox) test. A simple linear regression was performed for the correlation of monocytes with cytokines. The regression line was plotted and, in addition to the *p*-value, the coefficient of determination R^2^, which represents the measure of quality of the linear regression, was also provided. Odds ratio with 95% Confidence Interval (CI) was calculated using IBM SPSS Statistics 28.0.1 software (Armonk, NY, USA). Statistical significance was defined at *p*-values < 0.05. All *p*-values are two-tailed.

## 3. Results

### 3.1. Significantly Increased Levels of Monocytes in Peripheral Blood of PDAC Patients Are Linked to the Progression of the Disease and Mortality of the Patients

First, a comparison of the percentage of monocytes in the peripheral blood of PDAC patients to healthy individuals and pancreatitis patients was conducted. The gating strategy to distinguish monocytes in the peripheral blood is demonstrated in Figure 1A.

PDAC patients showed significantly higher levels of monocytes compared to healthy and pancreatitis patients, whereas no difference was observed between healthy individuals and patients diagnosed with pancreatitis (Figure 1B).

Next, the impact of circulating monocytes on the clinicopathological characteristics in PDAC patients was investigated. To this end, analysis of the lymph node status did not detect any difference between patients with invaded (pN^+^, *n* = 25) and spared lymph nodes (pN0, *n* = 15). With regard to perineural infiltration of the tumor, significantly higher levels of monocytes in the blood of patients diagnosed with infiltrating (Pn^+^, *n* = 24) compared to non-infiltrating (Pn0, *n* = 16) tumors were observed. While G3 tumors (*n* = 32) were associated with increased levels of circulating monocytes compared to G2 (*n* = 13) tumors, no changes were observed in respect to the tumors resection margin (R0, *n* = 37 vs. R^+^, *n* = 21) (Figure 1C).

As higher levels of monocytes were associated with more aggressive tumor growth, the role of circulating monocytes as prognostic biomarker for PDAC severity was investiagted. To identify the monocyte value that correlates with the highest probability of G3 and Pn^+^ PDAC occurrence, a minimal *p*-value approach (Table 2) was performed. Patients with monocytes >5% were more likely to develop a G3 and Pn^+^ PDAC. More specifically, patients with a monocyte percentage greater than 5% had an 8.9-fold increased risk of this more aggressive tumor (compared to patients with a monocyte percentage ≤ 5%) (OR: 8.889; 95% CI: 1.564–50.530) (Figure 1D).

To test the reliability of the newly implemented prognostic value, analysis of patient survival stratified by the monocyte percentage was performed. Patients with low levels of monocytes (≤5%, *n* = 15) showed significantly increased survival rates than patients with higher monocytes (>5%, *n* = 43) (Figure 1E). Characteristic features of the study population dichotomized by monocyte concentration are listed in Table 3.

### 3.2. Increased Expression of CD86, CD11c, and PD-L1 on Peripheral Blood Monocytes of PDAC Patients Siginificantly Correlates with Disease Severity

In the light of the aforementioned findings, the activation status of monocytes was analyzed in a prospective cohort (*n* = 26). A significantly increased expression of CD86 on circulating monocytes was found in comparison to healthy individuals. In addition, such CD86 expression was found to be higher in G3 compared to G2 tumors. With regard to the lymph node status (pN), the perineural invasion (Pn), and the resection status (R), no significant correlations were observed (Figure 2A).

CD11c was also expressed more abundantly on monocytes of PDAC patients compared to healthy individuals. Clinically, CD11c expression was shown to correlate with the invasion of lymph nodes. While the expression of CD11c was increased in patients with G3 and Pn^+^ tumors, the analysis of the resection status revealed no CD11c-related differences (Figure 2B).

PD-1 is known as a crucial receptor used by cancer to evade the immune system, with its ligand, PD-L1, being expressed by monocytes. Similarly, the expression of PD-L1 was lower on monocytes of PDAC patients when compared to healthy controls. Such decreased expression was also observed in patients with tumor-invaded lymph nodes (versus lymph node-spared patients). No significant correlations were noted between the PD-L1 expression and perineural invasion, tumor grading, and resection status (Figure 2C).

### 3.3. Frequencies of Circulating Monocytes Are Significantly Linked to Increased Concentrations of TNF-αand IL-6 in Plasma of PDAC Patients

Furthermore, the aim was to determine whether monocytes are associated with the most important proinflammatory and anti-inflammatory representatives of cytokines, as these may be involved in the tumor progression as well. The level of monocytes correlated positively with the concentrations of TNF-α (Figure 3A) and IL-6 (Figure 3B) in the plasma of PDAC patients, whereas no correlation to IL-10 (Figure 3C) was noted.

### 3.4. Increased Proportion of Intermediate Monocytes Is Linked to Disease Severity in PDAC Patients

Monocytes are known as heterogenous cells exhibiting ambigiuous activities during cancer disease. Therefore, the subsets of the circulating monocytes in PDAC patients were analyzed. The gating strategy applied to differentiate classical, non-classical and intermediate monocytes in the peripheral blood is demonstrated in Figure 4A.

The proportion of classical monocytes in the total monocyte population was found to be similar between PDAC patients and healthy individuals. When specifically analyzing classicial monocytes in PDAC patients, there were no significant differences with regard to histopathological and clinical characteristics (i.e., lymph node status, perineural invasion, tumor grading, and resection status) (Figure 4B). Likewise, the quantity of non-classicial monocytes did not differ between PDAC patients and healthy individuals.

In contrast, intermediate monocytes were found to be increased in PDAC patients compared to healthy individuals. While the lymph node status, the perineural tumor invasion, and the resection status remained comparable, G3 tumors were associated with a significantly higher percentage of intermediate monocytes compared to G2 tumors (Figure 4C).

Among the subset of non-classicial monocytes in PDAC patients no tumor characteristic-related differences could be identified (Figure 4D).

## 4. Discussion

We were able to show that PDAC patients had an increased percentage of circulating monocytes and that this was also associated with more aggressive tumor growth. The increased activation status of the monocytes suggests that these cells are, in fact, involved in tumor progression and that this finding was not only due to a decrease in other immune cells.

Monocytes are innate immune cells and originate from progenitors in the bone marrow [24]. While trafficking via the bloodstream monocytes can further differentiate into a range of tissue macrophages and dendritic cells [25]. In cancer, monocytes can exert both antitumoral toxicity and protumoral activity [26]. In this process, monocytes can differentiate into tumor-associated macrophages that promote tumor growth and metastasis [27]. Furthermore, a study of glioblastoma patients revealed that monocytes secreted the matrix-bound vascular endothelial growth factor (VEGF) and thus enhanced angiogenesis, which is crucial for tumor growth [28]. In PDAC, Sanford et al. demonstrated that circulating monocytes suppressed T-cell functions and recruited regulatory T-cells [22]. These findings support our hypothesis that monocyte analysis could serve as a reliable prognostic marker in PDAC.

PD-L1 is expressed in both monocytes in general and in a variety of malignancies. The binding to the inhibitory checkpoint molecule PD-1 promotes apoptosis of antigen-specific T-cells and reduces apoptosis in regulatory T-cells [29,30]. Monocytes are also capable of inducing natural killer cells to produce the anti-inflammatory interleukin-10 [31]. However, we found that the expression of PD-L1 on monocytes was decreased in PDAC patients compared to healthy individuals. This observation may have occurred because PD-L1 could already have interacted more frequently with PD-1 and thus promoted immune escape of the tumor.

Cytokines have been shown to be involved in tumor progression of various types of malignancies. In PDAC, TNF-α induces endothelial-mesenchymal transition promoting stromal development of the tumor [32], and therefore it is consistent that increased monocytes are associated with increased concentration of TNF-α and thus more aggressive tumor growth. However, the source of this cytokine in PDAC is not yet known and the intricacies of the relationship with monocytes in PDAC remains unexplored. IL-6 has a key role in PDAC development and progression: it effects immune suppression in the tumor microenvironment and enhances angiogenesis, proliferation and migration of tumor cells [33]. In a mouse model, the blockade of IL-6 and PDL-1 reduced tumor progression of PDAC [34]. The correlation of monocytes and IL-6 that we have shown supports our hypothesis that monocytes are associated with more aggressive tumor growth and might therefore serve as a possible prognostic biomarker in PDAC.

Three subtypes of monocytes can be distinguished. Classical monocytes are a type of inflammatory cells involved in host defence responses [35,36] whereas non-classical monocytes have a patrolling function and are involved in tissue repair and debris removal from the vasculature [37]. Interestingly, intermediate monocytes have a wide range of different functions: in addition to the production of reactive oxygen species (ROS), they can secrete either proinflammatory or anti-inflammatory cytokines [38,39]. In PDAC patients we were able to demonstrate that the proportion of intermediate monocytes was elevated and associated with higher grading of the tumor. This implicates their potential involvement in the tumor progression. However, further studies are necessary to clarify which mediators are predominantly secreted by intermediate monocytes in PDAC patients.

In various types of cancer including large B-cell lymphoma [40,41], cervical cancer [42], and stage III colon cancer [43], it has already been shown that elevated levels of circulating monocytes are associated with poorer survival rates. In ovarian cancer, it has been demonstrated that elevated monocytes, in addition to poorer survival rates, were also associated with more aggressive tumor growth [44]. The relationship between circulating monocytes and the clinical outcome in PDAC patients remained unexplored until now, but this present study was able to fill this gap.

The lymph node status, perineural invasion and the tumor grading are among the most important tumorbiologic prognostic factors in PDAC patients [45]; however, reliable biomarkers are still lacking to more accurately assess the response to adjuvant chemotherapy or identify patients at risk for recurrent disease, both of which directly impact patient outcome. The analysis of circulating monocytes as liquid biopsy could therefore close this gap and might be a valuable prognostic tool, since the examination can be performed more repeatedly without a large investment of time and resources. However, more studies are needed to establish actual use in the care of patients. Besides the prognostic value, it would be of interest to explore whether monocytes can also be harnessed as therapeutic targets in PDAC.

Nonetheless, some caution is advised concerning the findings of the present study. Due to the relatively low number of patients, the statistical power is limited. Besides that, only circulating monocytes were examined and no analysis of the tumor tissue was performed. Therefore, only assumptions can be made about the role of monocytes regarding tumor progression in PDAC patients.

## 5. Conclusions

We found monocytes to be significantly increased in PDAC patients. In addition, this abundance of monocytes correlated with the disease severity. As such, we herein propose monocytes as novel biomarker in PDAC. This finding may help surgeons to refine their serologic PDAC screening and optimize the perioperative care of PDAC patients. Future studies are needed to decipher the exact role of monocytes in PDAC. Such understanding of monocytes could be translated into individualized treatments based on biomarkers, with promising potential to improve the prognosis of this lethal malignancy.

## Figures and Tables

**Figure 1 cancers-15-00363-f001:**
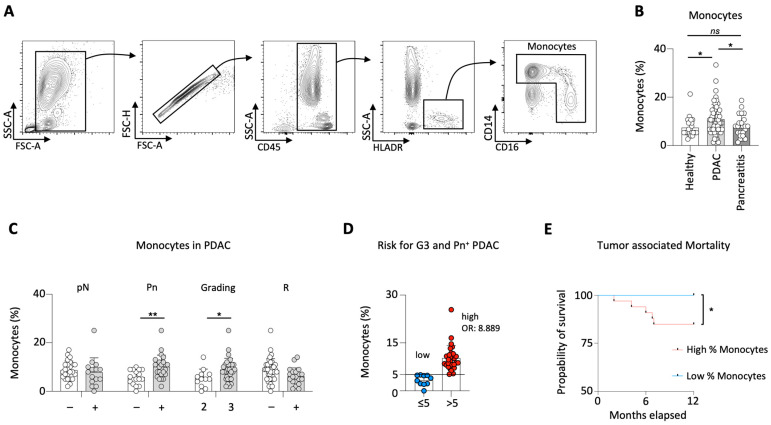
PDAC patients showed increased levels of monocytes. Gating strategy for circulating monocytes (**A**); percentage of circulating monocytes in healthy individuals, PDAC and pancreatitis patients (**B**); percentage of monocytes in PDAC patients correlated to the lymph node status (pN), perineural invasion (Pn), tumor grading (G) and the resection status (R) (**C**); percentage of monocytes (5% identified by minimal *p*-value approach) defining the risk for G3 and Pn^+^ PDAC patients (**D**); survival analysis for patients with high (>5%) vs. low (≤5%) monocytes (**E**); * *p* < 0.05; ** *p* < 0.01; *ns* no significance.

**Figure 2 cancers-15-00363-f002:**
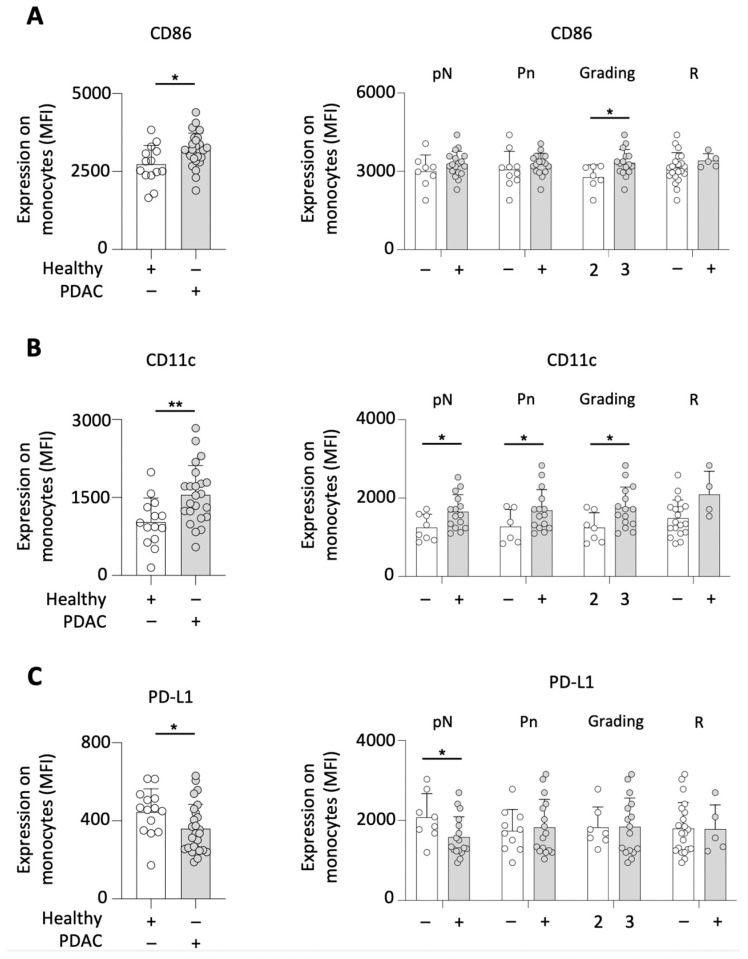
PDAC patients demonstrated an increased activation status of circulating monocytes. Analysis of PDAC patients’ monocytes compared to healthy individuals and correlation with the disease severity (lymph node status (pN), perineural invasion (Pn), the tumor grading (G) and the resection status (R)) for the expression of CD86 on (**A**); CD11c (**B**); and PD-L1 (**C**); * *p* < 0.05; ** *p* < 0.01.

**Figure 3 cancers-15-00363-f003:**
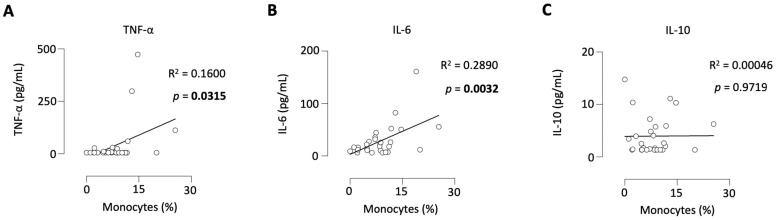
Levels of monocytes correlated positively with the TNF-α and IL-6 plasma concentrations in PDAC patients. Levels of monocytes correlated to the concentration of TNF-α (**A**), IL-6 (**B**), and IL-10 (**C**) in the plasma of PDAC patients.

**Figure 4 cancers-15-00363-f004:**
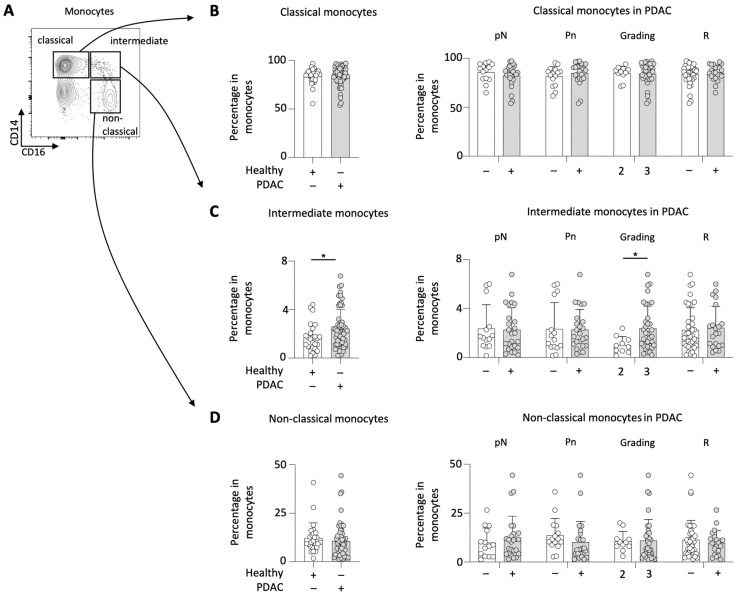
PDAC patients showed increased frequencies of intermediate monocytes. Gating strategy applied to characterize circulating monocyte subsets in PDAC patients (**A**), percentage of classical monocytes (**B**), intermediate monocytes (**C**), and non-classical monocytes (**D**) in PDAC patients compared to healthy controls and their correlation with the disease severity (lymph node status (pN), perineural invasion (Pn), the tumor grading (G) and the resection status (R)); * *p* < 0.05.

**Table 1 cancers-15-00363-t001:** Clinico-pathological parameters of the PDAC study cohort.

	PDAC Patients
Number	58
Mean Age (in years [range])	58 (42–92)
Sex (Male:Female)	31:27
Tumor size pT	
1	7
2	16
3	14
4	3
Unknown/Inoperable	18
pN-category	
pN0	15
pN^+^	25
Unknown/Inoperable	18
Perineural invasion	
Pn0	16
Pn^+^	24
Unknown/Inoperable	18
R-status	
R0	37
R^+^	21
Grading	
G2	13
G3	32
Unknown	13
Distant Metastasis	
No	41
Yes	17
UICC stage	
I	2
II	15
III	6
IV	17
Unknown/Inoperable	18
Neoadjuvant treatment	
Radiochemotherapy	4
Chemotherapy	9
-	45

**Table 2 cancers-15-00363-t002:** Determining the cutoff threshold of the percentage of monocytes based on the aggressiveness of tumor growth (G3 and Pn^+^) using the two-tailed minimal *p*-value approach (chi-square test; *n* = 39). The optimal cutoff value with the lowest *p*-value is marked grey. Statistically significant valus are highlighted in bold.

Monocytes (%)	*p*-Value (chi-sqare Test)	Monocytes	N	G3 and Pn^+^
**>3**	0.18	Low High	4 35	25.0% 60.0%
**>4**	**0.03**	Low High	6 33	16.7% 63.7%
**>5**	**<0.01**	Low High	10 29	20.0% 69.0%
**>6**	**0.02**	Low High	12 27	33.3% 66.7%
**>7**	**0.02**	Low High	13 26	30.8% 69.2%
**>8**	0.09	Low High	17 22	41.2% 68.2%
**>9**	**0.02**	Low High	24 15	41.7% 80.0%

**Table 3 cancers-15-00363-t003:** Clinico-pathological parameters of the PDAC study cohort stratified by the percentage of monocytes in low (*n* = 15; ≤5%) and high (*n* = 43; >5%).

Percentage of Monocytes		Low	High	*p*-Value
Number		15	43	
Mean Age (in years [range])		69 (51–90)	64 (45–86)	0.7068
Sex	Male (%)	6 (40)	25 (58)	0.23
	Female (%)	9 (60)	18 (42)	
Tumor size pT	1 (%)	3 (20)	4 (9)	0.40
	2 (%)	2 (13)	13 (30)	
	3 (%)	5 (33)	9 (21)	
	4 (%)	0 (0)	3 (7)	
	Unknown/Inoperable (%)	5 (33)	14 (33)	
pN-category	pN0 (%)	6 (40)	8 (19)	0.18
	pN^+^ (%)	4 (27)	21 (49)	
	Unknown/Inoperable (%)	5 (33)	14 (32)	
Perineural invasion	Pn0 (%)	8 (53)	7 (16)	**<0.01**
	Pn^+^ (%)	2 (13)	22 (51)	
	Unknown/Inoperable (%)	5 (33)	14 (33)	
R-status	R0 (%)	9 (60)	30 (70)	0.49
	R^+^ (%)	6 (40)	13 (30)	
Grading	G2 (%)	8 (53)	5 (12)	**<0.01**
	G3 (%)	5 (33)	27 (63)	
	Unknown (%)	2 (13)	11 (26)	
Distant Metastasis	No (%)	9 (60)	32 (74)	0.29
	Yes (%)	6 (40)	11 (26)	
UICC stage	I (%)	2 (13)	5 (12)	0.75
	II (%)	4 (27)	18 (42)	
	III (%)	1 (7)	5 (12)	
	IV (%)	6 (40)	11 (26)	
	Unknown/Inoperable (%)	2 (13)	4 (9)	
Neoadjuvant treatment	No (%)	10 (67)	35 (81)	0.24
	Yes (%)	5 (33)	8 (19)	

## Data Availability

For all data requests, please contact the corresponding author.

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
