# Peer review of "Circulating Monocytes Serve as Novel Prognostic Biomarker in Pancreatic Ductal Adenocarcinoma Patients"

_cancers, 2023, doi:10.3390/cancers15020363_

Round 1
Reviewer 1 Report
Authors described novel biomarker “monocyte” for PDAC. Elevated monocytes are associated with prognosis in PDAC patients. I read this paper with interest and think it is a good study. Below are some questions I have and some points I would like to confirm.
Minor:
In Table 3, I think it would be easier to understand if there are percentages next to each number.
I think PD-L1 is more common, not PDL-1.
Authors said about subtypes of monocytes, “This implicates their potential involvement in the tumor progression.”. Since monocytes are essentially immune cells, how about the interpretation that as the malignancy progresses, monocytes increase not to help the cancer, but to fight the cancer cells?
There is a word "identiy" that I think is simply misspelled.
It is a more immature study, but a search on PubMed for “monocyte” and “PDAC” shows a report on monocyte elevation in 2019. What is the relevance to your study?
DOI: 10.1016/j.pan.2019.10.002
Author Response
Reply to Reviewer 1
We thank this reviewer for the constructive comments.
Comments:
Authors described novel biomarker “monocyte” for PDAC. Elevated monocytes are associated with prognosis in PDAC patients. I read this paper with interest and think it is a good study. Below are some questions I have and some points I would like to confirm.
- In Table 3, I think it would be easier to understand if there are percentages next to each number.
Thank you very much for this comment. We have added the percentage to the table.
- I think PD-L1 is more common, not PDL-1.
Thank you very much for noticing that. We have adjusted the manuscript accordingly.
- Authors said about subtypes of monocytes, “This implicates their potential involvement in the tumor progression.”. Since monocytes are essentially immune cells, how about the interpretation that as the malignancy progresses, monocytes increase not to help the cancer, but to fight the cancer cells?
Thank you very much for this interesting comment. Monocytes and their subtypes have both pro- and anti-tumoral functions. When searching the literature, pro-tumoral discoveries of monocytes in cancer predominate. Should a subtype of monocytes have a protective effect in PDAC, there possibly would be a significant decrease in the clinical correlation between healthy controls and PDAC patients. However, we cannot answer the question conclusively, as a more detailed analysis of the subtypes would be necessary. Unfortunately, we do not have a panel in which we can analyze activation markers or PD-L1 for monocyte subtypes. Therefore, further studies are necessary to answer this very interesting question.
- There is a word "identiy" that I think is simply misspelled.
Thank you for this comment. We have rectified the same.
- It is a more immature study, but a search on PubMed for “monocyte” and “PDAC” shows a report on monocyte elevation in 2019. What is the relevance to your study?
Thank you for discovering this study. This study confirms our finding that monocytes are increased in PDAC patients. However, we could additionally show that monocytes are associated with more aggressive cancer growth. In addition, we were able to determine an exact statistical value that could serve as a prognostic factor. Thus, we bring our discoveries closer to clinical use. Furthermore, we could show an increased activation status of monocytes in PDAC patients and the positive correlation with IL-6 and TNF-alpha.
Reviewer 2 Report
Thanks for the opportunity to review the manuscript "Circulating Monocytes Serve as Novel Prognostic Biomarker in Pancreatic Ductal Adenocarcinoma Patients" by Hansen et al. This analysis deals with the findings of circulating monocytes in PDAC patients. Blood samples from PDAC patients, patients with pancreatitis and healthy humans were collected and analysed. The authors report about a significance increase of circulated monocytes in PDAC patients and propose monocytes as novel biomarker in these patients. Some negative aspects stand out during the reading of the manuscript.
Methods: Authors reported about samples sizes collected from 58 PDAC and 24 pancreatitis patients, the healthy control group ist not named in the methods section. Please add the healthy population. Table 1 seems to be part of the result section.
Results: In a scientific text the use of "we" is not recommended in the result section (we compeared, we investigated, ..). This reduces the scientific message content that is present in the text!
I would recommend that your manuscript should undergo extensive scientific english revisions.
Author Response
Reply to Reviewer 2
We thank this reviewer for the constructive comments.
Comments:
Thanks for the opportunity to review the manuscript "Circulating Monocytes Serve as Novel Prognostic Biomarker in Pancreatic Ductal Adenocarcinoma Patients" by Hansen et al. This analysis deals with the findings of circulating monocytes in PDAC patients. Blood samples from PDAC patients, patients with pancreatitis and healthy humans were collected and analysed. The authors report about a significance increase of circulated monocytes in PDAC patients and propose monocytes as novel biomarker in these patients. Some negative aspects stand out during the reading of the manuscript.
- Methods: Authors reported about samples sizes collected from 58 PDAC and 24 pancreatitis patients, the healthy control group is not named in the methods section. Please add the healthy population. Table 1 seems to be part of the result section.
Thank you very much for noticing that. We have added the control group in the Materials and Methods section.
- Results: In a scientific text the use of "we" is not recommended in the result section (we compeared, we investigated, ..). This reduces the scientific message content that is present in the text!
We thank the reviewer for pointing this out. We have changed all the `we` into a content which sounds more scientific.
- I would recommend that your manuscript should undergo extensive scientific english revisions.
Thank you for this suggestion. We revised the manuscript accordingly.